# In Silico Substrate-Binding Profiling for SARS-CoV-2 Main Protease (M^pro^) Using Hexapeptide Substrates

**DOI:** 10.3390/v15071480

**Published:** 2023-06-29

**Authors:** Sophakama Zabo, Kevin Alan Lobb

**Affiliations:** Department of Chemistry, Rhodes University, Makhanda 6139, South Africa; sophakamazabo@gmail.com

**Keywords:** SARS-CoV-2 main protease, protein substrate, multi-conformer substrate library, molecular docking, molecular dynamics, PCA

## Abstract

The SARS-CoV-2 main protease (M^pro^) is essential for the life cycle of the COVID-19 virus. It cleaves the two polyproteins at 11 positions to generate mature proteins for virion formation. The cleavage site on these polyproteins is known to be Leu-Gln↓(Ser/Ala/Gly). A range of hexapeptides that follow the known sequence for recognition and cleavage was constructed using RDKit libraries and complexed with the crystal structure of M^pro^ (PDB ID 6XHM) through extensive molecular docking calculations. A subset of 131 of these complexes underwent 20 ns molecular dynamics simulations. The analyses of the trajectories from molecular dynamics included principal component analysis (PCA), and a method to compare PCA plots from separate trajectories was developed in terms of encoding PCA progression during the simulations. The hexapeptides formed stable complexes as expected, with reproducible molecular docking of the substrates given the extensiveness of the procedure. Only Lys-Leu-Gln*** (KLQ***) sequence complexes were studied for molecular dynamics. In this subset of complexes, the PCA analysis identified four classifications of protein motions across these sequences. KLQ*** complexes illustrated the effect of changes in substrate on the active site, with implications for understanding the substrate recognition of M^pro^ and informing the development of small molecule inhibitors.

## 1. Introduction

Since its emergence in late 2019, the death toll due to Coronavirus disease (COVID-19) has reached over 6.9 million people, with more than 760 million cases recorded globally according to the WHO [1]. COVID-19 generally manifests as a pulmonary disease caused by Severe Acute Respiratory Syndrome Coronavirus 2 (SARS-CoV-2) infection [2], but it also has fatal extrapulmonary manifestations [3,4]. SARS-CoV-2 belongs to a diverse family of enveloped viruses consisting of large, single-stranded, positive-sense RNA genomes of around 27–32 kb in length—typically composed of a 5′-methylguanosine cap at the beginning, a 3′-poly-A tail at the end, and a total of 6–10 genes in between [5]. The viral genome has high frequencies of genomic recombinations and mutations, as demonstrated by the existence of its variants [6,7]. The SARS-CoV-2 genome encodes at least 27 proteins, including 16 non-structural proteins (nsp1–10, nsp12–16), 4 structural proteins, and 8 accessory proteins (ORF3a, ORF3b, ORF6, ORF7a, ORF7b, ORF8, ORF9b, and ORF14) [3,8].

Among the 16 encoded nsps is the main protease (M^pro^), also referred to as chymotrypsin-like protease (3CLpro), which is a homodimeric, 33.8 kDa cysteine protease encoded as nsp5 by the open reading frame OFR1a/b [9,10]. Each protomer of the main protease consists of catalytic dyads (HIS41 and CYS145), which cleave the polyproteins into mature proteins translated by ORF1a and ORF1ab. The main protease is first autocleaved from the polyprotein (pp1ab) and then downstream nsps (nsp4-nsp16) are cleaved at 11 different sites of the pp1ab—recognizing the sequence Leu-Gln↓(Ser/Ala/Gly) (↓ shows the cleavage site) [6,11]. This protein is vital to the life cycle of coronaviruses, as it directly mediates the maturation of the nsps essential for viral replication [6].

With the emergence of variants with increased virulence and advanced evasive mechanisms against host immunological defences [12,13,14], existing pharmacological measures have lost their efficacy, necessitating the development of broad-spectrum chemotherapies. There are no vaccines that confer long-term immunity and consistently elicit immunological protection against a broad range of SARS-CoV-2 variants. Optimal pharmacological measures to control the spread and SARS-CoV-2-related morbidity have yet to be discovered and developed. Simulations on the cleavage points of pp1ab with proteases show interesting changes in stability depending on the protease complex [15]. Hence, in this study, we explored SARS-CoV-2 M^pro^ proteolytic substrate recognition as a viable technique to gain an insight into the molecular cleavage of pp1ab substrates at the active site. It is hoped that this in turn will rationally inform the development of anti-coronavirus compounds. 

Viral proteases have proven to be excellent drug targets that have led to the development of effective drugs against chronic infections. The functional importance of the main protease makes it a prudent target for antiviral drug discovery against SARS-CoV-2, as its inhibition could potentially hinder the viral replication cycle, and overall, stall the production of infectious SARS-CoV-2 virions [6,10]. Importantly, the structural architecture of the main protease is highly conserved across various coronaviruses, despite the extensive mutagenesis undergone by coronaviruses [11]. Therefore, the development of main protease inhibitors could lead to the creation of broad-spectrum antiviral therapeutic agents against SARS-CoV-2 and other coronaviruses, while reducing the risk of mutation-mediated drug resistance in future deadly viral strains [6,11]. Moreover, there are no human proteases with analogous substrate specificity to the protein and therefore, main protease inhibitors are more likely to be harmless to patients [11]. The present work hypothesized profiling the substrate binding at the catalytic site to understand the function and mechanisms of SARS-CoV-2 M^pro^. As per the hypothesis, a multi-conformer hexapeptide library—following the recognition and cleavage sequences for M^pro^—were generated and used to evaluate binding efficiency and substrate recognition and specificity against the crystallographic structure of M^pro^. Accordingly, the behavior of M^pro^ in the presence of the hexapeptides was assessed using molecular dynamics simulations.

## 2. Materials and Methods

### 2.1. Creation of Multi-Conformer Hexapeptide Library

Due to the unavailability of oligopeptide libraries consisting of peptide chains with SARS-CoV-2 M^pro^ recognition sequences, a hexapeptide (P3-P3′) library was thus generated on a local server using RDKit (v. 2019.09.1) within the Python scripting environment. The constituent amino acid residues were selected from the findings of Ullrich and Nitsche [10], prioritizing high-occurrence amino acids in the consensus sequence over all cleavage sites. The peptide structures were constructed using the MolFromSequence method and subsequently capped with acetyl (ACE) and methylamine (NME) constructs at the C- and N- termini, respectively. Hydrogen atoms were removed and added to ensure each atom valency was satisfied. The three-dimensional (3D) coordinates of the atoms were generated using the EmbedMolecule method and the molecules were optimized using the implemented UFFOptimizeMolecule method. A structural conformational search was performed to generate 100 conformers per system. The resulting conformers were stored in an SDF file.

### 2.2. Molecular Docking

To construct the 3D M^pro^–hexapeptide complexed structures, molecular docking simulations were performed using AutoDock Vina [16]. The crystallographic 3D structure of SARS-CoV-2 M^pro^ was retrieved from the RCSB Protein Data Bank (PDB) (PDB id: 6XHM). All of the co-crystallized ligands and waters were removed. The hexapeptide structures were subjected to geometry optimization employing a semi-empirical quantum mechanical method (XTB semi-empirical method) using xTB software [17] and converted to PDB format using openBabel [18]. Gasteiger charges and atom types were assigned to the receptor, and the peptide substrates were prepared using Python scripts provided by AutoDockTools [19]. Preliminary molecular docking simulations resulted in favorable binding to protomer B over protomer A (Appendix A). Therefore, a grid box size was set at 20 × 20 × 20 Å and centered at −18.444, −16.361, and 7.944 in the x, y, and z directions, respectively, targeting the protomer B-binding pocket of the receptor. The energy range and exhaustiveness were set at 4 and 480, respectively. The docking simulations were performed in parallel using 24 central processing unit (CPU) cores per job at the Centre for High Performance Computing (CHPC). After the hexapeptides were docked into the binding pocket, the ligand PDBQT files were split into separate files and the poses with the lowest docking scores were prioritized for further study. The validation and reproducibility of the docking results were confirmed for the best poses per hexapeptide by means of structural superimposition, subsite mapping, and intermolecular interactions, which were explored using PyMOL [20] and BIOVIA Discovery Studio 2020 Client [21]. Respective M^pro^–hexapeptide complexed structures were constructed using conformers with the best pose.

### 2.3. Molecular Dynamics Simulations

For the molecular dynamics simulations, a focused subset of the 810 hexapeptides was used. The 20 ns molecular dynamics (MD) simulations were performed using GROMACS (version 2018.1) for SARS-CoV-2 M^pro^ in the absence and presence of hexapeptides containing the sequence KLQ***, for a total of 131 complexes. Together with the *apo* system, this accounted for 132 MD simulations. Substrates were combined as a third protein chain in the receptor pdb file. The topology and coordinate files for the *apo*–M^pro^ and M^pro^–hexapeptide systems were created using the AMBER03 protein, nucleic AMBER94 force field [22]. The 132 systems were solvated in a cubic box of 10 nm dimension using the TIP3P water model and subsequently neutralized with 0.15 M NaCl. The solvated systems were minimized for 50,000 steps using the steepest descent minimization algorithm until maximum forces of <10.0 kJ/mol were achieved. After minimization, the systems were equilibrated with an NVT ensemble (constant number of atoms, volume, and temperature) at 300 K for 100 ps, followed by an NPT ensemble (constant number of atoms, pressure, and temperature) equilibration at 1.0 bar for 100 ps. A modified Berendsen thermostat algorithm was used for both equilibration steps. For production runs, the time steps were set at 2 fs, whereas the trajectory and coordinate information were saved every 10 ps, resulting in 2000 frames saved for every system simulated. The LINCS holonomic constraints algorithm was deployed to constrain all bonds, whereas the Particle Mesh Ewald algorithm was set for all long-range electrostatics. The entire procedure was automated on the CHPC cluster, using 8 nodes per job with 24 cores per node and 24 processes per node. Following completion of production runs, the periodic boundary conditions were removed and the trajectories were analyzed according to root mean square deviation (RMSD), root mean square fluctuation (RMSF), radius of gyration (Rg) and principal component analysis (PCA).

### 2.4. Principal Component Analysis and Protein Motion Classification

Conformational changes and structural protein motions over the course of the trajectory for the protein backbone were monitored using PCA. Due to the high volume of PCA data, a custom pairwise comparison of the molecular dynamics simulations was conducted in an attempt to classify the similarities and differences in M^pro^ during the 132 simulations. The Cartesian coordinates of PC1 and PC2 for each system were fitted in a 5 × 5 grid to demarcate the protein motions as defined by PCA over the course of the simulation. In intervals of 2 ns (200 frames), the PC1 and PC2 coordinates of the protein were averaged and subsequently used to position the PCA within the grid. These averaged figures were assigned a letter code corresponding to the position within the grid of the PCA in the particular time interval. The resulting string of letter codes (for the full simulation time) thus tracked the progression of the PCA for the respective simulation. The differences in these codes between separate molecular dynamics simulations were used to create a pairwise comparison that included all 132 PCA analyses, under the assumption that the protein motions described by PC1 and PC2 s were the same in the compared systems. The differences between these PCA components were clustered using correlation as a measure of distance. Hierarchical clustering was performed to indicate the hierarchical relationships in the PCA progression with regard to dynamic protein motions, thereby grouping all systems in terms of similar molecular dynamics. The molecular dynamics of two systems from each class/group were visually inspected using Visual Molecular Dynamics (VMD) [23] to verify protein motion similarity and dissimilarity in appropriate pairs of the 132 PCA plots.

## 3. Results and Discussion

### 3.1. The Multi-Conformer Hexapeptide Library

A total of 810 hexapeptides were generated from the high-occurrence amino acids in the consensus sequence over all cleavage sites derived from Ullrich and Nitsche [10]. Using the information from Ullrich and Nitsche [10], octapeptides were initially considered, which would have resulted in 61,000 possible substrates. That many substrates would have required computational power and resources beyond the feasibility of our study. Hexapeptides, on the other hand, provided the appropriate computational manageability and intensity to study the substrate-binding profiles of SARS-CoV-2 M^pro^. Since only the P3-P1’ residues are required for recognition, specificity, and cleavage, working with hexapeptides (the P3-P3′) provided the largest set of substrates that was still computationally manageable. The resulting library consisted of peptide chains demonstrating physicochemical diversity that allowed for broad-spectrum characterization of M^pro^–substrate interaction (Table 1). The combinations of high-occurrence amino acids used to create the multi-conformer library allowed for the assessment and study of potential/promising substrates that are not necessarily naturally occurring within the SARS-CoV-2 genome yet still adhere to the substrate recognition and specificity of M^pro^; most importantly, they possessed desirable properties that could serve as a rational basis for drug design and discovery. By doing so, this study incorporated wide-range characterization of M^pro^ functionality (keeping in mind its genomic and structural conservation across variants and species), and in turn, provides insight into the protein’s behavior in the presence of mutant and homologous forms of the SARS-CoV genome and polyprotein substrates. Such information could serve as a basis for the development of broad-spectrum chemotherapies that retain efficacy in spite of emerging mutations.

Following the construction of each hexapeptide, terminal capping was performed to enhance the structural stability of the peptide substrates, particularly in dynamic environments. The subsequent conformational search provided an opportunity to identify the most favorable geometries prior to binding. This could have future use in the design of peptidomimetics displaying potent antiviral activity. Ultimately, docking studies were performed with 81,000 systems, comprising 100 conformers for each of the 810 hexapeptides.

### 3.2. Virtual High-Throughput Screening of the Hexapeptides

A quality crystal structure of SARS-CoV-2 M^pro^ was retrieved from the RCSB PDB (PDB ID: 6XHM) showing good resolution of 1.41 Å and no mutations or missing residues. The protein was reported as encoded using the SARS-CoV-2 MN908947.3 genome and synthesized via *Escherichia coli* expression [24]. Moreover, the crystal structure provided three rotamers of M^pro^ and this study utilized the rotamer designated “A”.

The high-throughput screening of the hexapeptides was performed on protomer B targeting the active site pocket, after preliminary docking studies assessing both protomers (Appendix A). Overall, the best docked poses (or docked pose with minimum energy) registered binding free energies ranging between −8.7 and −7.0 kcal.mol^−1^ across all 810 substrates. The mean binding free energies of the hexapeptides—grouped by P3-P1 residues—alongside their respective ligand efficiencies are reported in Table 2. These docking results were arguably good and alluded to high-affinity binding of the substrates onto the active site of SARS-CoV-2 M^pro^, as to be expected. The ligand efficiencies indicated that all of the P3-P1 residue combinations displayed similar goodness of interaction with the receptor and thus showed similar potential improvement in binding affinity for structural- and efficiency-driven drug design [25,26,27]. Therefore, all of these P3-P1 residue combinations suggested a viable basis for characterization of the active site for rational antiviral drug design, given their ligand efficiency (Table 2).

Further evaluation and validation of the docking results revealed the reproducibility of the best docking pose together with conducive binding interactions with the active site residues for these particular poses. The visualization of the best poses of conformers per hexapeptide showed overlap in the backbone (α-carbons) of the peptide substrates when superimposed, with only a few substrates displaying alternative poses with minimum energy (Appendix A). This was an impressive feat considering the flexible nature of the peptide macromolecules. Figure 1A shows the superimposed poses with minimum binding free energy occupying the same docking/binding space. The use of a high degree of search exhaustiveness (of 480) substantially increased the probability of finding plausible poses that were reproducible.

Mapping of the substrates in the binding pocket of M^pro^ was used to visually assess the substrate recognition in accordance with the nomenclature of Schechter and Berger [28] and Ullrich and Nitsche [10]. Ullrich and Nitsche [10] stipulated that SARS-CoV-2 M^pro^ mainly recognizes substrate residues P4-P1′; however, substrate specificity is determined by residues P2-P1′, as demonstrated by the highest degree of conservation across the pp1a/ab cleavage sites. The substrate residues were recognized and anchored within the binding pocket by specific active site residues that comprised subsites. Table 3 lists these subsites as well as their constituent active site residues. These subsites surrounded the catalytic dyad consisting of His41 and Cys145, which mediates digestion of the polyproteins [10]. The key active site residues that essentially mediate substrate binding and processing include His41, Met49, Gly143, Ser144, His163, His164, Met165, Glu166, Leu167, Asp187, Arg188, Gln189, Thr190, Ala191, and Gln192 [11].

Figure 1B shows the binding mode of the hexapeptide within the binding pocket with respect to the active site subsites [11,29,30]. The P3-P1 residues were mounted desirably onto the active site, with each residue being anchored onto its appropriate subsite. The desired binding modes and adherence of the hexapeptide binding modes to the nomenclature of Schechter and Berger [28] were prevalent across the best docked poses. Violation of the nomenclature of Schechter and Berger [28] was only apparent in some of the top docked poses. Hexapeptides such as RLQATF and RLQSTF—the top poses (−8.7 kcal.mol^−1^)—showed the side chain of P3 anchored in S1, whereas the side chains of P2 and P1 were anchored in S3 and S2, respectively. Others, such as RLQAAN, showed S1 anchoring the side chains of P1 and P3, whilst S2 rightfully anchored P2 (Appendix A). Nonetheless, most binding modes showed that the M^pro^ crystal structure followed the substrate recognition commonly associated with SARS-CoV-2 M^pro^. The side chains of the hexapeptides were appropriately accommodated in the subsites of the active site, regardless of significant differences in docking scores. Furthermore, these modes confirmed that these hexapeptides, as informed by the literature, were recognized by M^pro^ and appropriately bound to the receptor protein. Thus, substrate recognition by M^pro^ towards the generated hexapeptides was confirmed. Ultimately, the establishment of this recognition for the hexapeptides will assist in efforts to design potent and therapeutic peptidomimetics.

Resolution of the intermolecular interactions at the binding interface showed the high prevalence of hydrogen bonds and van der Waals interactions forming across all hexapeptides and active site residues, as shown in Figure 1C. The formed hydrogen bonds included conventional hydrogen bonds, carbon–hydrogen bonds, and Pi-donor hydrogen bonds. Hydrogen bonds, especially conventional hydrogen bonds, are the fundamental stabilizing force in biomolecular structures that underpin structure, function, and conformational dynamics [31,32], and they also regulate complementarity and stability in protein–ligand complexes at the binding interface [33,34]. Therefore, the prevalence of hydrogen bond formation at the active site indicated shape and electrostatic complementarities between M^pro^ and the hexapeptides, and it also pointed to high affinity for the peptides, as evidently shown by the binding free energies and desirable binding modes (Figure 1B–D; Appendix A). Furthermore, the stabilizing effect of hydrogen bonds would prove to be integral in maintaining the complex structures in dynamic processes.

Notably, the key active site residues (including the catalytic dyad) frequently participated in hydrogen bonds and van der Waals interactions with the hexapeptides. Cys145 typically formed conventional hydrogen bonds with the oxygen atoms of the carboxyl group of P1, which in turn placed the catalytic residue in close proximity to the scissile peptide bond. Key residues 138 to 146, which constitute the oxyanion loop that confers substrate stability during the proteolytic process, participated in key stabilizing forces (hydrogen bonds and van der Waals forces of attraction) that promoted the formation of stable complexes. Other key residues, such as Met49, His163, His164, Met165, Glu166, Leu167, Asp187, Arg188, Gln189, Thr190, Ala191, and Gln192, which underpinned the subsites and accommodated the appropriate binding of the substrate residues via side-chain rearrangement [11,35,36,37], also formed the aforementioned stabilizing interactions with the hexapeptides. Figure 1D shows that the entirety of the hexapeptide was anchored by various hydrogen-bonding interactions formed with active site residues. Conclusively, stable M^pro^–hexapeptide complexed structures were constructed as shown by many stabilizing interactions at the binding interface. Moreover, the prevalent formation of hydrogen bonds at the binding interface pointed to the substrate specificity that a coronavirus main protease would typically exhibit towards peptide structures with the Leu-Gln↓(Ala/Ser) cleavage sequence.

### 3.3. Global Stability of the M^pro^ Systems

The 20 ns MD simulations were performed on 131 Lys-Leu-Gln*** (KLQ***)–M^pro^ complexed structures and an unbound M^pro^ (*apo*-) structure. The exclusive use of KLQ*** complexed structures for molecular dynamics was to preserve comparability between systems—maintaining that the ligands shared identical P3-P1 residues and that they held similar ligand efficiencies with other P3-P1 ligand groups in this study. The global stability of the M^pro^ systems was assessed through the calculation of root mean square deviation (RMSD) and radius of gyration (Rg) values for the protein backbone (α-carbons) of M^pro^ based on the 20 ns MD simulation trajectories, which were plotted as violin plots to highlight significant changes in conformation and stability (Figure 2 and Figure 3).

The RMSD plots (Figure 2) showed the various behaviors of the protein across the systems and mainly displayed three distinct trends in structural stability listed in the following order of prevalence: substantial fluctuation, stable conformation, and significant changes in location (dissociation). Varying degrees of backbone fluctuation and conformational shifts were prevalent among the KLQ*** systems, as illustrated by bimodal and multimodal clustering and clustering at high RMSD values (>0.2 nm). Bimodal clustering, as clearly depicted in the KLQAGV, KLQAVE, KLQSAA, and KLQSLA systems, suggested that the protein converged on two alternative stable conformations during the simulation—an indication of reduced overall stability. Multimodal clustering (KLQAAA, KLQAEA, KLQAGM, KLQAGQ, KLQAKE, KLQAKN, etc.) and clustering at high values (KLQAED, KLQAGA, KLQAGD, KLQSKA, and KLQSKD) indicated substantial fluctuation of the backbone as the protein backbone struggled to achieve favorable conformations with the ligands, thus leading to overall conformational instability.

The *apo*-system and several other KLQ*** complexed systems displayed stable conformations, to varying degrees, as shown by the unimodal distribution of RMSD values. The unimodal distribution indicated a single dominant conformation for each M^pro^ system. The *apo*-system together with the KLQAAG, KLQAEG, KLQAEM, and KLQAKQ systems and others achieved moderate stability throughout the simulation. The plots suggested that these systems attained equilibration around 0.2 nm, as indicated by clustering, with a mean RMSD value of around 0.2 nm. The best stability of the M^pro^ backbone was demonstrated by the KLQAAD, KLQAKA, KLQASG, KLQAVD, KLQSAV, KLQSND, and KLQSNQ systems, all which achieved unimodal clustering at values less than 0.2 nm. In essence, these systems achieved favorable conformations with their ligands with low backbone fluctuations.

Conversely, significant changes in conformation and/or location in the M^pro^ structure were shown in the KLQAEQ, KLQAND, KLQSGA, and KLQSVQ systems, as they registered significantly steep changes in RMSD values during their respective simulations. The plots for the KLQAEQ and KLQSGA systems indicated the dissociation of the hexapeptide from the binding pocket, whereas the plots for the KLQAND and KLQSVQ systems showed the dissociation of the M^pro^ protomers.

The behaviors of M^pro^ in the context of compactness and folding followed similar trends as the backbone fluctuations (RMSD values) for Rg values (Figure 3). However, stable compactness was more prevalent than structural unfolding and/or stretching from the protein’s center of mass. Unimodal distribution of the Rg values was achieved by the majority of M^pro^ systems, wherein stable compactness was typically attained roughly around 2.56 to 2.60 nm. The KLQAAD system attained the lowest stable compactness, around 2.57 nm. Other stable systems included the unbound (*apo*), KLQAEE, KLQAGA, KLQAKA, KLQAKG, KLQAKV, KLQASG, KLQATD, KLQAVG, KLQSAG, KLQSAN, KLQSEE, KLQSEM, KLQSGE, KLQSKA, KLQSND, KLQSNN, KLQSTG, KLQSVA, KLQSVD, KLQSVE, and KLQSVM, all of which maintained a stable state below 2.60 nm.

The unfolding and/or stretching of the protein backbone during the simulations were shown to varying degrees, which were indicated in the Rg plots as bimodal clustering, stretched bimodal clustering, and little to no clustering (stretched distribution) (Figure 3). Similar to the RMSD values, the M^pro^ systems such as KLQASV, KLQATE, and KLQSAM registered bimodal clustering, suggesting that the protein backbone converged into two stable states. The stretched bimodal clustering in the Rg plots indicated that the M^pro^ backbone converged into multiple conformations but briefly attained two stable states, as depicted in the KLQAAV, KLQAED, KLQAGV, KLQALD, KLQASD, KLQSAA, KLQSAQ, KLQSEG, KLQSKD, and KLQSKV systems. The absence of clustering in other systems, such as KLQAAA, KLQAAE, KLQATE, KLQASV, KLQAVE, KLQAVQ, KLQSKM, KLQSNA, and KLQSNM, was indicative of steady increases in Rg values as a consequence of gradual stretching of the M^pro^ backbone during the simulation. This lack of clustering could indicate minor unfolding of the protein in the presence of the respective hexapeptides. However, as previously noted, the stretching of the M^pro^ backbone was gradual and not attributed to drastic denaturing of the M^pro^ structure. Lastly, the KLQAEQ, KLQAND, KLQSGA and KLQSVQ systems registered steep hikes in Rg values similar to the RMSD values. These steep changes in Rg values were due to the aforementioned dissociation events displayed in the RMSD plots. Appendix A provide the exact values for RMSD and Rg in terms of mean and standard deviation for all of the molecular dynamics simulations.

### 3.4. Local Stability of the M^pro^ Systems

Local chain fluctuations of M^pro^ were measured by computing the root mean square fluctuation (RMSF) and assessed using heatmaps. Heatmaps allowed the identification of high-flexibility regions, and their subsequent mapping on the M^pro^ crystal structure revealed the positions of these regions within the 3D protein structure (Figure 4, Figure 5 and Figure 6). Across all M^pro^ systems, the RMSF values of both protomers of the protein approximated each other, with protomer A registering slightly higher RMSF values despite having no bound hexapeptide (Figure 4A,B; Figure 5A,B; Figure 6A,B). Protomer B is where the bound hexapeptide resided. Overall, the highest RMSF values were obtained in the KLQAND and KLQSVQ systems (Figure 6) due to the dissociation of the protomers. Since these values were disproportionately higher than those of the rest of the systems, the data were separated to optimize visualization as follows: (i) systems with KLQ*** substrates with Ala at P1′ (Figure 4); (ii) systems KLQ*** substrates with Ser at P1′ (Figure 5); and (iii) KLQAND and KLQSVQ systems (Figure 6).

Figure 4 shows the heatmaps for both protomers of the Lys-Leu-Gln-Ala** (KLQA**) systems alongside *apo*-M^pro^. While the RMSF values slightly differed from one system to the next, both protomers of M^pro^ displayed high flexibility in the same regions in all systems (Figure 4A,B). Overall, the RMSF values for the KLQA** systems ranged between 0.0384 and 0.5805 nm for protomer A and 0.0394–0.5698 nm for protomer B. High flexibility was observed in residues 21–26, 44–80, 92–97, 118–127, 141–144, 152–156, 167–171, 188–198, 215–288, and 298–302 in protomer A (Figure 4A) and residues 1–4, 22–24, 44–80, 92–96, 118–125, 153–156, 168–171, 188–197, 212–288, and 297–301 in protomer B (Figure 4B), respectively. The Lys-Leu-Gln-Ser** (KLQS**) systems showed similar trends as the KLQA** systems in terms of RMSF values and the localization of flexible residues (Figure 5A,B). The overall range of RMSF values was between 0.0399 and 0.4223 nm for protomer A and 0.0372–0.3726 nm for protomer B. Similarly, flexibility was demonstrated in residues 21–26, 32–35, 44–80, 92–97, 119–123, 139–143, 153–156, 167–171, 187–197, 212–288, and 297–302 in protomer A and residues 1–5, 21–26, 33–35, 44–66, 70–80, 92–98, 152–156, 167–170, 186–197, 212–238, 241–286, and 297–301 for protomer B.

The majority of the highly fluctuating residues constituted loop regions in both protomers of M^pro^, as shown (as red, and blue) in Figure 4C and Figure 5C. This high fluctuation could be attributed the flexible nature of loop structures. However, of all the flexible loop regions, residues 1–4 displayed flexibility exclusively in protomer B. Residues 1–9 comprise the N-finger terminal region, which plays a crucial role in dimer formation through interactions with Domain II of protomer A [38]. Semi-flexibility in β-sheets (Domains I and II) was displayed on either ends of the structure in both protomers—connecting to or from loop regions. In addition, α-helices displayed semi-flexibility in residues 44–80 (domain I) and 212–288 (domain III). Of note, residues 44–80 are part of the catalytic domain responsible for catalysis and M^pro^ autocleavage [30]. Sequentially, this α-helix comes after catalytic His41 and consists of key binding residues such as Met49, which contributes to substrate stabilization. While the RMSD and Rg values did not indicate overall destabilization of the M^pro^ structure, this apparent semi-flexibility could be attributed to functional flexibility and the intrinsic side-chain rearrangement mechanisms that accommodate the hexapeptide in the binding pocket. This point was further supported by the absence of hexapeptide ejection from the binding pocket. The α-helices of Domain III (Figure 4C and Figure 5C; the lower region of protein) consistently displayed high flexibility in both protomers. The high fluctuation could be attributed to their connection to long loop regions, as α-helices typically demonstrate restricted protein motion but can confer great flexibility, which is essential to protein function [39].

In the context of the final two systems, KLQAND and KLQSVQ—for which the residue fluctuations were the highest among all systems due to dimer dissociation—RMSF values ranging between 0.2261 and 1.1565 nm for protomer A and 0.2174–1.1333 nm for protomer B were registered. However, the flexible residues were similar to those in the KLQA** and KLQS** systems, including residues 1–17, 69–73, 96–100, 111–127, 138–144, 152–157, 202–209, 210–223, 224–234, 236–237, 242–254, 255–259, 260–276, 277–285, 286–298, and 299–302 in protomer A and residues 1–19, 24–29, 69–74, 95–100, 111–128, 138–143, 151–157, 170–173, 199–206, 207–223, 224–227, 247–288, 291–299, and 300–301 in protomer B. Notably, these were the only instances where the N-finger terminal residues displayed flexibility in both protomers (Figure 6), as expected, since dimer dissociation occurred in both of these systems.

Furthermore, the localization of the flexible residues showed greater residue fluctuation in Domain II involving entire β-sheet structures (Figure 6C). This was not evident in the other systems, as fluctuations in β-sheets were only registered for their terminal ends connecting to loop regions. The semi-flexibility in α-helices (residues 44–63) surrounding the catalytic dyad was absent in these two systems, pointing to reduced activity and/or inactivity of the active site residues. Moreover, the α-helix residues constituting Domain III demonstrated much greater flexibility than any of the other systems, further indicating destabilization of the M^pro^ structure.

In conclusion, the RMSF results supported that the M^pro^ residues typically displayed similar fluctuation patterns in the presence of the KLQ*** substrates, with the exception of the KLQAND and KLQSVQ systems. The RMSF values of both protomers approximated one another in magnitude, and the locations of the high-fluctuation regions on the M^pro^ protomers were similar. Interestingly, the KLQAEQ and KLQSGA systems did not register high RMSF values like the KLQAND and KLQSVQ systems, but instead behaved similar to the majority of complexed systems. Therefore the unbinding of the hexapeptides did not cause significant changes to the motions and conformations of M^pro^ in these two systems (KLQAEQ and KLQSGA). Consequently, the steep hikes in RMSD and Rg values were only attributed to the fact the MD program interpreted the hexapeptide chain as a third chain in the overall protein system.

### 3.5. Principal Component Analysis and Protein Motion Classification

The prominent structural motions and conformational changes of the M^pro^ backbone during the 20 ns MD simulations were assessed using PCA calculations. PCA divided the overall protein motions of the trajectories into principal components that described the essential functional protein motions during the simulation. Since the first two principal components, PC1 and PC2, retained the majority of the variance of the original data, they could be used to provide a meaningful description of the protein motions throughout the course of the simulations. Thus, 2D projections of these principal components were plotted using the Cartesian coordinates of all backbone atoms to visualize and examine these conformational changes (Appendix A). The direction of change for the movement of M^pro^ against time was unique to each system. For example, some systems displayed clockwise changes in PCA, while others showed anti-clockwise changes in PCA. Seemingly, almost all systems retained steady conformational changes throughout the simulations as the distribution of the PCA coordinates was generally compact/clustered. Without regard to their pattern of progression, the typical range of the coordinates was between −5 and 5 for both PC1 and PC2. The KLQAEQ, KLQAND, KLQSGA, and KLQSVQ systems displayed the most drastic changes in protein motions due to the aforementioned changes in stability, structure, and conformation.

In an attempt to classify these PCA data, a custom pairwise comparison of the M^pro^ systems was performed as detailed in Section 2.4. Figure 7 shows the cluster map illustrating the correlations of the protein motions for all systems with respect to one another (bearing in mind the underlying assumption). In the cluster map, systems sharing similarities in protein motions were registered in hues between black and violet, whereas systems with dissimilarities were registered in hues between white and orange. The accompanying dendrogram illustrates the similarities or correlations among the prominent protein motions of the clustered M^pro^ systems. Hierarchical clustering yielded four main classifications of correlation among all the systems that depicted similarity in proteins motions that describe the trajectories of the M^pro^ systems. The four classifications of similarity in the M^pro^ systems are listed in Table 4.

Although three classifications of the systems shared varying similarities within and between them, the Group 1 PCA data (comprising the KLQSVQ, KLQAEQ, and KLQAND systems) demonstrated the highest dissimilarity to the rest of the PCA groups, as expected, considering their patterns in RMSD, Rg, and even RMSF values (for the KLQAND and KLQSVQ systems). The PCA of the KLQAND and KLQAEQ systems showed high similarity, as opposed to that of the KLQSVQ system. The KLQAND and KLQSVQ systems both showed dimer dissociation, although not similarly, during their simulations. Nevertheless, the clustering was successful in identifying highly dissimilar protein motions in the M^pro^ systems, as mutually confirmed by the RMSD, Rg, and RMSF data. Interestingly, the KLQSGA system, which also showed structural instability in its RMSD and Rg values, was classified in Group 3 and not alongside the KLQSVQ, KLQAEQ, and KLQAND systems. Of note, the dendrogram showed the KLQSGA system displaying the highest dissimilarity with the rest of the systems within Group 3, showing a weak correlation in terms of protein motions and conformational changes with its “similar” systems.

Systems in Groups 2 to 4 all showed varying degrees of similarity within and across themselves. Systems in Group 3 displayed the least amount of divergence in the hierarchical dendrogram when compared to Groups 2 and 4 (Figure 7). Notably, Group 3 was occupied by systems that attained high medians and/or multimodal distribution in their RMSD and Rg values. Systems that demonstrated stability were split across the three groups.

The PCA results highly aligned with the RMSD, Rg, and RMSF values in terms of overall trends. Most of the M^pro^ systems registered PCA coordinates within the range of 5 and −5 for PC1 and PC2, just as they attained RMSD and Rg equilibration around 0.2 nm and 2.60 nm, respectively. Importantly, there were only two unbinding events registered to the 131 KLQ***-bound M^pro^ systems—an indication of strong intermolecular interaction in the binding pocket. The retention of hexapeptides by the protein could be attributed to the substrate recognition and specificity postulated in the docking studies. Despite binding substrates with minor chemical differences, the protein achieved and retained similar flexibility, compaction, and prominent protein motions, which altogether caused similar behaviors and patterns of stability in the M^pro^ systems. Consequently, these results suggested that SARS-CoV-2 M^pro^ has an intrinsic mechanism that enables the binding of different peptide substrates without rendering drastic instability to the entire structure. Even in the PCA, the majority of the systems seemingly occupied the same positions on the PCA plot throughout the course of the simulation. The exceptions to these trends consistently included the KLQAEQ, KLQAND, KLQSGA, and KLQSVQ systems, as clearly demonstrated by their high dissimilarity to the rest of the systems. While the dissimilarities in protein motions depicted in the dendrogram (Figure 7) could be easily explained, the similarities also required verification. Henceforth, at least two relevant systems from each hierarchical classification (or group) were selected for trajectory visualization using VMD.

The visualization of the trajectories for the KLQAEQ and KLQSVQ systems in Group 1 confirmed the weak correlation in protein motions reported in the dendrogram. The protein backbone for both systems showed no overlap at any point in the simulations (Appendix A). Dissimilarity in these systems was expected as the KLQAEQ system showed substrate unbinding while the KLQSVQ system showed dimer dissociation. Therefore, these systems only fell within the same classification due to their drastic changes in conformation and location, as shown by their RMSD and Rg values. While the ejection of non-covalently bound ligands from the binding pocket normally does not register such extreme changes in RMSD and Rg values for the protein backbone, the steep hikes displayed in the KLQAEQ system were attributed to the inclusion of substrates as third chains of M^pro^ by the GROMACS software prior to the MD simulations. The protein motions of the hexapeptide when it was ejected and later bound again affected the values for RMSD, Rg, and PCA. The KLQAND system seemingly displayed similarity to the KLQAEQ system, even though the former demonstrated dimer dissociation. The visualization of these systems refuted this similarity as the protein backbones did not exhibit overlap during the simulations (Appendix A). However, the relationship between their PCA results could be attributed to the substrate unbinding (KLQAEQ) and dissociation (KLQAND) that were later restored for both systems. Therefore, the similarity between the PCA results was owing to both systems having registered steep hikes in RMSD and Rg values rather than steep gradients in RMSD and Rg values.

The visualization of the systems (*apo*-M^pro^ and M^pro^-KLQSEG) in Group 2 showed a strong similarity in protein motions as the structures overlapped throughout all frames of the trajectories (Appendix A). Particularly, Domains I and II (the chymotrypsin-like structure) consistently overlapped throughout the simulation, whereas the helices of Domain III displayed the most structural deviation in both structures. The mapping of high-RMSF-value residues indicated that residues in this domain were highly flexible. The KLQAVV and KLQSGA systems in Group 3 displayed little similarity in protein motions at any point of the simulations (Appendix A). This was supported by the placement of these systems in the dendrogram; wherein the KLQSGA system exhibited high dissimilarity with the rest of the systems in Group 3 owing to the unbinding of the hexapeptide towards the end of the simulation. However, the similarity between the KLQSGA system and other Group 3 systems could have been the result of overlapping M^pro^ motions prior to unbinding of the hexapeptide. The KLQAVV system demonstrated structural stability as well as ligand stability throughout the simulation and also showed conformational similarity with the closely related KLQSAN system (Appendix A).

Similarly, the systems (M^pro^-KLQAAA and M^pro^-KLQSKG) in Group 4 also showed little similarity in protein motions (Appendix A). While overlapping of the structures was visible in the β-sheets of Domain I, a strong similarity was highly unlikely considering their placement in the dendrogram (Figure 7). It could be, in general, that the protein motions were within a broad range, but it was visually impossible to assess this detail. Lastly, the visualization of the systems from all four groups revealed little to no similarity in protein motions between groups (Appendix A). Structural overlaps rarely occurred during the simulations. The dissimilarity in protein motions was highly expected given the fact that these classifications diverged from the highest point of dissimilarity, as illustrated in Figure 7. Thus, this observation validated the hierarchical clustering since these systems exhibited no similarity in their prominent protein motions throughout the 20 ns MD simulation.

The visualization of the four clades of PCA progression allowed the inspection and confirmation of the similarities in protein motions of the systems. The outcomes of the visualization indicated that this type of PCA progression analysis is able, certainly in some cases, to identify both similar and dissimilar protein motions in comparable systems. With refinement, it could provide information for general use. In this case, it certainly enabled us to focus on particular systems given the number of simulations to assess.

## 4. Conclusions

COVID-19, as a disease and pandemic, continues to pose a tremendous threat to global stability as it has the potential to strain public health services to extraordinary proportions, cause economic disruptions and/or inactivity leading to financial crises, and predominantly sabotage the livelihood of millions of people around the world. Moreover, the existing measures in place to control the spread of the virus are losing efficiency due to the emergence of novel variants with increased virulence and immunological evasive mechanisms. SARS-CoV-2 M^pro^ has proven to be a promising drug target as it exhibits high degrees of conservation in sequence, structure, and specificity. However, there are details of the mechanism of proteolysis employed by SARS-CoV-2 M^pro^ that still require study. Thus, this study was carried out to profile the binding of hexapeptide substrates onto SARS-CoV-2 M^pro^ in preparation for the elucidation of its proteolytic mechanism and, ultimately, rational substrate-based and efficiency-driven drug design to combat SARS-CoV-2 and its variants. In summary, the virtual high-throughput screening of the hexapeptides showed favorable binding in terms of binding free energy score, ligand efficiency, as well as substrate recognition and specificity. Based on the MD simulations performed, M^pro^ displayed various conformational behaviors but overall persistent stability in all complexed systems, with the exception of four outliers. The PCA showed compact distribution of PC1 and PC2 in all but the same four systems. The custom pairwise comparison for the quantification of PCA progression revealed four main classifications of similarity in protein motions, which were confirmed by visualization of the trajectories.

## Figures and Tables

**Figure 1 viruses-15-01480-f001:**
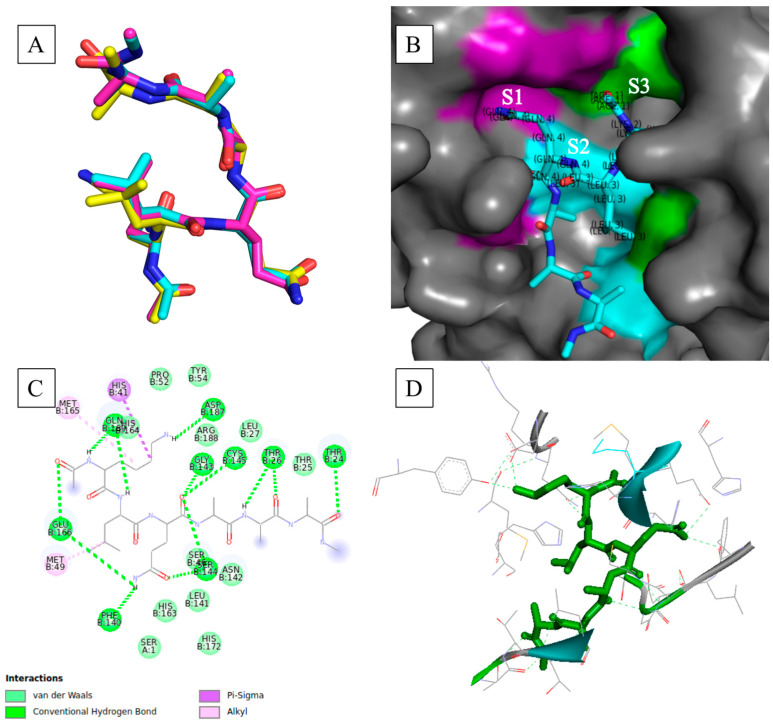
The assessment and validation of docking results for the SARS-CoV-2 main protease. (**A**) Validation of the reproducibility of the docking results. The visualization of the best poses of substrate Lys-Leu-Gln-Ala-Ala-Ala. Image was generated using PyMOL. (**B**) The surface of SARS-CoV-2 M^pro^ (PDB ID:6XHM) showing docked substrate and the substrate-binding subsites are color-coded as follows: purple: S1, cyan: S2; green: S3. The image was generated using PyMOL. (**C**) Resolution of intermolecular interactions between M^pro^ and substrate at the active site. Two-dimensional representation of the protein–ligand interactions at the active site for M^pro^ complexed with Lys-Leu-Gln-Ala-Ala-Ala substrate. The image was generated using BIOVIA Discovery Studio 2020 Client. (**D**) Hydrogen anchoring of the best docked pose. Three-dimensional representation of the Lys-Leu-Gln-Ala-Ala-Ala substrate residues forming hydrogen interaction with active site residues of M^pro^. The Lys-Leu-Gln-Ala-Ala-Ala substrate attained a binding free energy of −8.3 kcal.mol^−1^. The image was generated using BIOVIA Discovery Studio 2020 Client.

**Figure 2 viruses-15-01480-f002:**
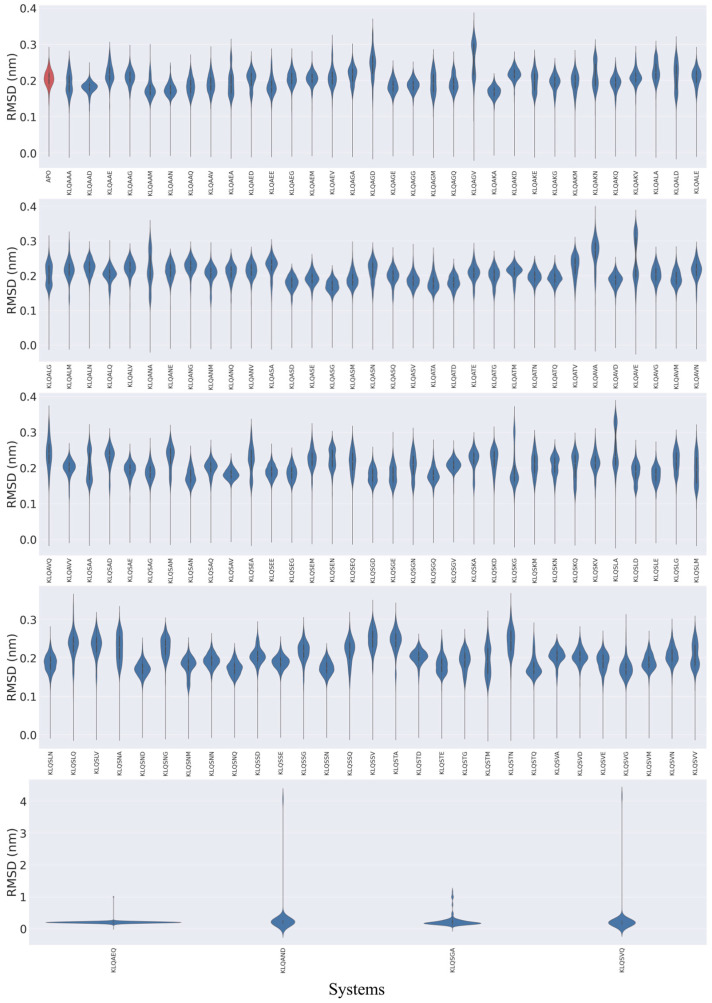
The violin plots of M^pro^ RMSD values in unbound (red) and hexapeptide-bound (blue) systems during the 20 ns MD simulation. Images were generated using Seaborn on Python.

**Figure 3 viruses-15-01480-f003:**
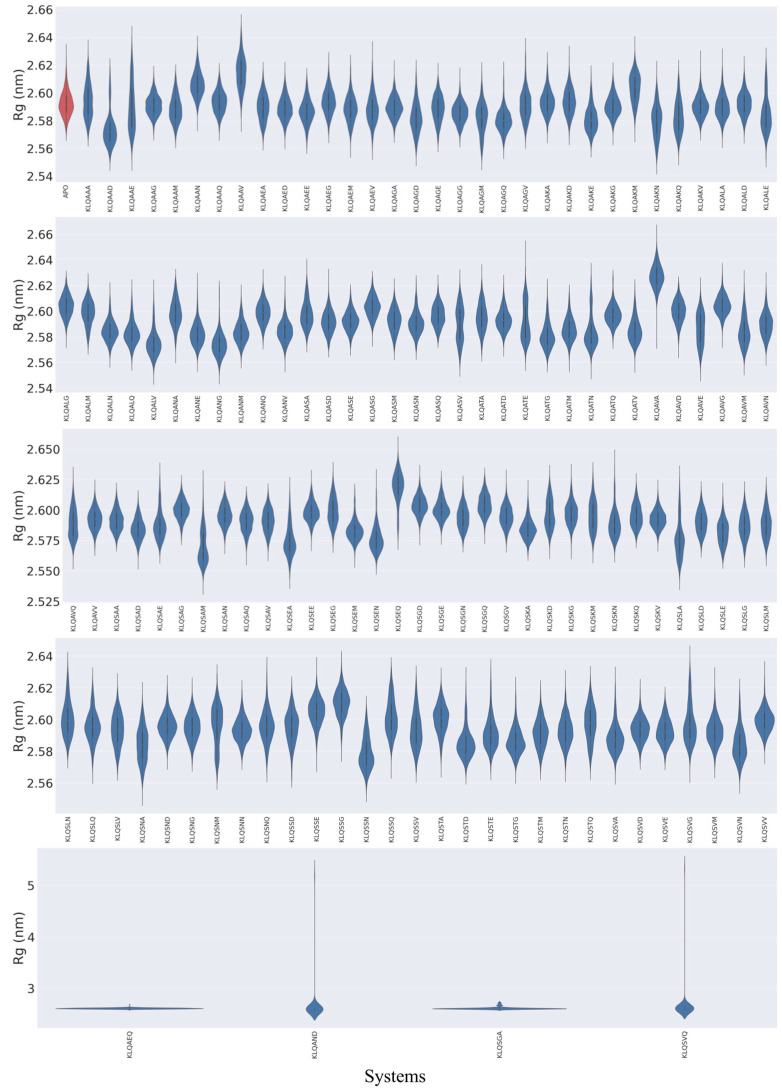
The violin plots of M^pro^ Rg values in unbound (red) and hexapeptide-bound (blue) systems during the 20 ns MD simulation. Images were generated using Seaborn on Python.

**Figure 4 viruses-15-01480-f004:**
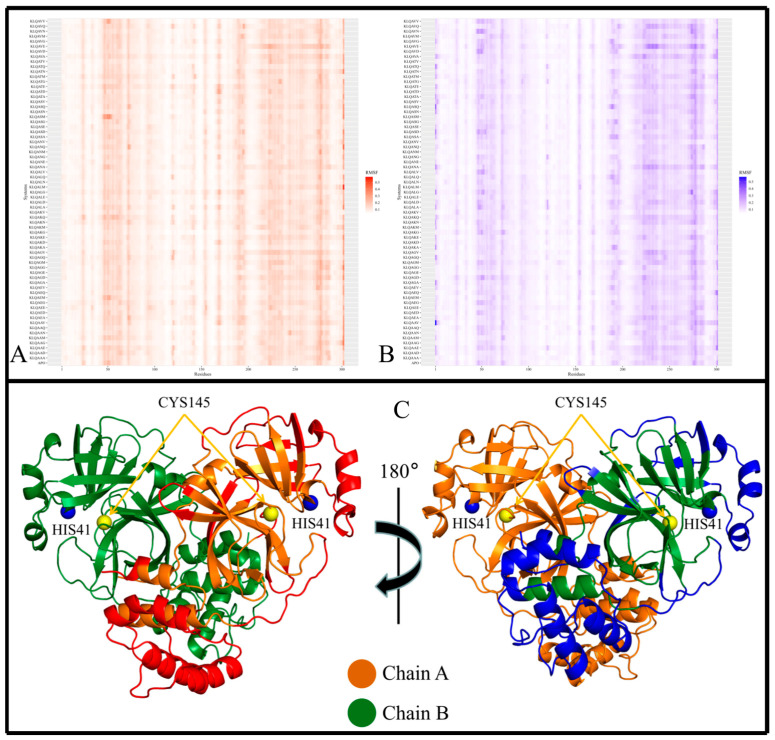
The local stability of M^pro^ and M^pro^–Lys-Leu-Gln-Ala** (KLQA**) hexapeptide complexes. RMSF values of the backbone α-carbon atoms for the *apo*-protein and KLQA** hexapeptide-bound M^pro^ systems during the 20 ns MD simulation. (**A**) The heatmap distribution of RMSF values for protomer A. (**B**) The heatmap distribution of RMSF values for protomer B. (**C**) The crystal structure of M^pro^ with regions of consistent higher RMSF values colored in red for protomer A and in blue for protomer B. Images were created using RStudio and PyMOL.

**Figure 5 viruses-15-01480-f005:**
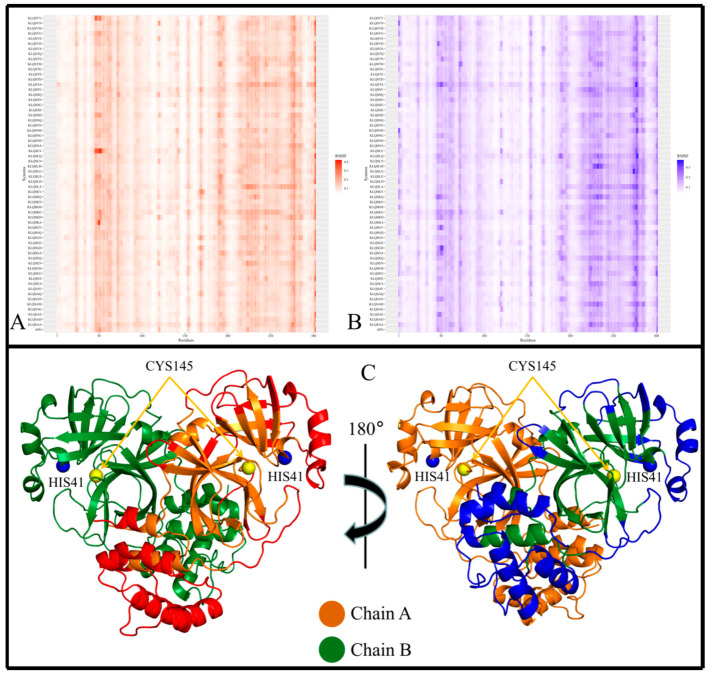
The local stability of the M^pro^ and M^pro^–Lys-Leu-Gln-Ser** (KLQS**) hexapeptide complexes. The RMSF values of the backbone α-carbon atoms for the *apo*-protein and KLQS** hexapeptide-bound M^pro^ systems during the 20 ns MD simulation. (**A**) The heatmap distribution of RMSF values for protomer A. (**B**) The heatmap distribution of RMSF values for protomer B. (**C**) The crystal structure of M^pro^ with regions of consistent higher RMSF values colored in red for protomer A and in blue for protomer B. Images were created using RStudio and PyMOL.

**Figure 6 viruses-15-01480-f006:**
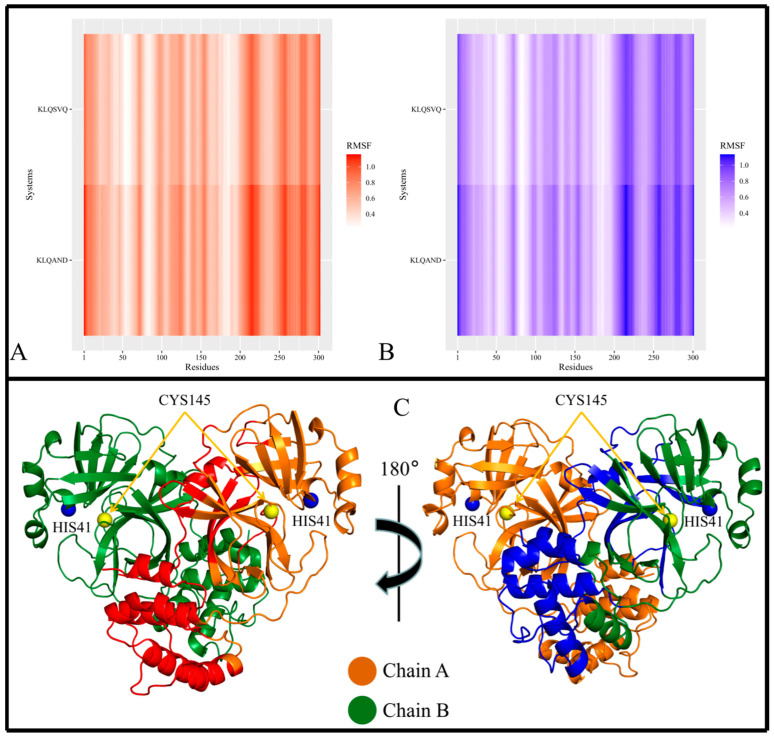
The local stability of the M^pro^–KLQ*** hexapeptide complexes. RMSF values of the backbone α-carbon atoms for the KLQAND and KLQSVQ hexapeptide-bound M^pro^ systems during the 20 ns MD simulation. (**A**,**B**) Refer to the protomers of the M^pro^ homodimer. (**C**) Refers to the crystal structure of M^pro^ with regions of consistent higher RMSF values colored in red for protomer A and in blue for protomer B. Images were created using RStudio and PyMOL.

**Figure 7 viruses-15-01480-f007:**
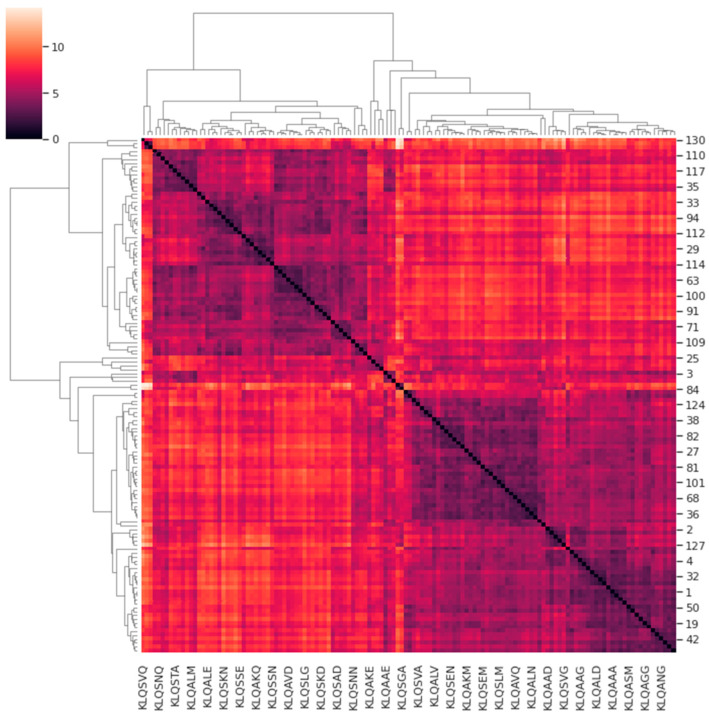
The correlations and similarities in protein motions of the M^pro^ dynamic systems. The clustering of differences in protein motions of the M^pro^ systems using correlation as a metric. The dendrogram illustrates hierarchical clustering of dynamic protein motions in the *apo*-M^pro^ and M^pro^–hexapeptide systems as a measure of similarity. The figure was generated using Seaborn in Python.

**Table 1 viruses-15-01480-t001:** The amino acid residues in the polyprotein cleavage sites recognized by SARS-CoV-2 M^pro^ used in the construction of hexapeptides. The amino acids are colored following the coloring scheme of WebLogo *.

Residue Position	Amino Acids
P3	**Arg**; **Lys**; **Met**; **Thr**; **Val**
P2	**Leu**
P1	**Gln**
P1′	**Ala**; **Ser**
P2′	**Ala**; **Asn**; **Glu**; **Gly**; **Leu**; **Ser**; **Thr**; **Val**
P3′	**Ala**; **Asn**; **Asp**; **Gln**; **Glu**; **Gly**; **Met**; **Phe**; **Val**

* Green: polar; Blue: basic; Red: acidic; Black: hydrophobic; Purple: neutral.

**Table 2 viruses-15-01480-t002:** The docking scores and ligand efficiencies of the hexapeptide substrates docked onto SARS-CoV-2 M^pro^ on the basis of P3-P1 variable amino acid residues.

Sequence (P3-P1)	Docking Score (kcal.mol^−1^)	HA *	LE ** (kcal.mol^−1^/HA)
Lys-Leu-Gln	−8.0 ± 0.26	28	−0.3 ± 0.01
Met-Leu-Gln	−7.9 ± 0.30	27	−0.3 ± 0.01
Arg-Leu-Gln	−7.9 ± 0.33	30	−0.3 ± 0.01
Thr-Leu-Gln	−7.9 ± 0.26	26	−0.3 ± 0.01
Val-Leu-Gln	−8.0 ± 0.23	26	−0.3 ± 0.01

* HA: heavy atoms (non-hydrogen atoms); ** LE: ligand efficiency.

**Table 3 viruses-15-01480-t003:** The active site residues constituting the substrate-binding subsites.

Subsites	Active Site Residues	Function	References
S3	His41; Met49; Met165	Shallow subsite that tolerates different functionalities.	[6,29]
S2	Thr25; His41; Met49; Cys145	Deeply buried subsite involved in hydrophobic and electrostatic interactions.	[6,30]
S1	Phe140; Gly143; Cys145; His163; Glu166; His172	Deeply buried subsite involved in hydrophobic and electrostatic interactions.	[6,30]
S1′	Thr25; Thr26; Leu27; Cys145	Generally forms polar contact interactions with substrates.	[30]

**Table 4 viruses-15-01480-t004:** The similarities among the dynamic M^pro^ systems on the basis of prominent protein motions and conformational changes during the 20 ns MD simulation.

	Group 1	Group 2	Group 3	Group 4
M^pro^ Systems	KLQSVQ; KLQAEQ; KLQAND	KLQANV; KLQSNN; KLQAKE; *APO*; KLQSEG; KLQSAG; KLQSAV; KLQSSV; KLQALG; KLQATD; KLQSGE; KLQSNQ; KLQAAM; KLQAEA; KLQSKN; KLQSLG; KLQSLV; KLQSSE; KLQALA; KLQSAA; KLQSGD; KLQSKD; KLQSTD; KLQAGE; KLQAKV; KLQATM; KLQAVG; KLQAKG; KLQAVD; KLQATV; KLQSTE; KLQSTG; KLQAKA; KLQSLQ; KLQSTN; KLQSAD; KLQASN; KLQSSD; KLQALE; KLQAED; KLQSGN; KLQAKQ; KLQASA; KLQATE; KLQAGD; KLQSSN; KLQSVE; KLQSAM; KLQSKA	KLQSGA; KLQSLE; KLQAAE; KLQAGQ; KLQSAN; KLQALM; KLQSVD; KLQASQ; KLQAAQ; KLQAEM; KLQSEQ; KLQAVV; KLQSTA	KLQAAG; KLQSGV; KLQANA; KLQAKN; KLQAAA; KLQSEA; KLQAVA; KLQAAD; KLQALQ; KLQSVG; KLQSAE; KLQSAQ; KLQAGM; KLQSNA; KLQSVV; KLQANG; KLQSGQ; KLQSLN; KLQALD; KLQAGV; KLQAKD; KLQATA; KLQAGG; KLQASV; KLQSKV; KLQANM; KLQAVN; KLQSKG; KLQSTQ; KLQSLA; KLQSNM; KLQSND; KLQASM; KLQSTM; KLQANQ; KLQSLD; KLQSEE; KLQSEM; KLQSLM; KLQSVN; KLQAEE; KLQAGA; KLQAVM; KLQSNG; KLQATQ; KLQSSG; KLQALN; KLQAEG; KLQATG; KLQATN; KLQAVQ; KLQASG; KLQAAV; KLQASD; KLQSKM; KLQAVE; KLQSEN; KLQASE; KLQALV; KLQANE; KLQAEV; KLQSVA; KLQAAN; KLQAKM; KLQSKQ; KLQSSQ; KLQSVM

## Data Availability

Data is contained within the article or Appendix A.

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
