# Peer review of "In Silico Substrate-Binding Profiling for SARS-CoV-2 Main Protease (M^pro^) Using Hexapeptide Substrates"

_viruses, 2023, doi:10.3390/v15071480_

Round 1
Reviewer 1 Report
In this manuscript, the authors study the substrate recognition of SARS-CoV-2 main protease (Mpro) by utilizing candidate hexapeptides and then filtering their chemistry through extensive virtual screening and molecular dynamics (MD) simulations. They find that hexapeptides beginning with the KLQ motif are especially stable and present further analysis on what the structural and functional space of such peptides look like in the context of binding to Mpro.
I am curious why the authors chose hexa-peptides, ie. what motivated the choice of 6 amino acid ligands. Did they perform experiments to detect the minimum peptide length required for its action as a substrate? Please make this clear in the text.
Prior to the AutoDock screening, the authors generate hexapeptide conformations using RDKit and parameterize the charges using Gasteiger charges. It is not clear if they continued to use these charges in the Gromacs topology file later, during the MD simulations. Gasteiger charges are extremely inaccurate for MD simulations, and typically the am1-bcc level of quantum theory is used to derive partial charges that are more consistent with Amber-94 and similar forcefields. Can the author’s make sure that the partial charge parameterization while setting up the MD simulations is consistent with the Amber-94 forcefield.
It is unclear to me, why the protomer-B active site on the Mpro was selected as the epitope for docking. Please provide more explanation for site-selection here.
In Table-2, please don’t call it delta-G, since it is not a true free energy. Instead, just say docking score.
For the global stability measurements of the ligand bound Mpro system in Fig 3.3, did the authors compute the RMSD of the entire Mpro or just around the active site?
While the authors develop new methods for calculating a pairwise correlation between dynamics PCAs of separate protein structures, I’m curious why they chose PCAs for this task, and that too moving. Why not pick something simpler like the average radius of gyration (Rg) ?
In a related question. Rg values are not really used in the analysis, so why did the authors compute them at all?
Reviewer 2 Report
In this manuscript, Zabo and Lobb reported their computational results of hexapeptide substrates interacting with the main protease (Mpro) of SARS-CoV-2. The hexapeptide sequences from P3 to P3’ were generated on the basis of the published papers. There were 810 hexapeptide substrates created for virtual high-throughput screening. Among all these substrates, authors found the best docking scores were in the range of −8.7 and −7.0 kcal.mol−1. The P3−P1 amino acid residues with a sequence of KLQ, MLQ, RLQ, TLQ, and VLQ provided high-affinity binding to the active sites of the SARS-CoV-2 Mpro, which are consistent with the published papers. Then authors took the substrates with the sequence of KLQ*** (i.e., Lys-Leu-Gln-***) and performed molecular dynamic simulations. The aim of this manuscript is to study the binding profile of hexapeptide substrates on the active sites of SARS-CoV-2 Mpro.
The SARS-CoV-2 Mpro has been recognized as a promising target in the development of antivirals for COVID. Although this manuscript reported a lot of molecular docking results, its main drawback is that the computational experiments did not include a published main protease inhibitor as a reference to conform their hypothesis. It also lacks of biological data, such as inhibition constant (Ki) and anti-viral activity (EC50), to support their computational results.
Other critical points indicated below need to be revised and improved.
1. The background of their hexapeptide substrate design was not described clearly in the Introduction. In other words, it is not persuasive and needs more literatures and rationale to explain why hexapeptides were chosen by the authors.
2. Authors should use 3-letter abbreviation instead of 1-letter abbreviation, such as Lys, Leu, and Gln for K, L, Q, respectively, for the discussion on molecular docking results and ligand efficiency (i.e., Table 2 and Figure 1). The reason is that the side chains of the amino acid residues are of importance to be investigated for the interaction between the substrates and the active sites of the enzymes.
3. There were over hundreds of complexes calculated in this manuscript. Authors may have to summarize the results of “3.3 Global stability of the Mpro systems” and “3.4 Local stability of the Mpro systems” into one or two tables. Their outcomes will then be easily understood by readers. As a result, their binding profile based on the structural characteristics of the amino acid residues can be drawn point-by-point.
4. Some parts in the Abstract and Conclusion are suggested to rewrite. For example, four different types of motion of complexes (line 18), small changes of substrate (line 19), except for four (line 617), four main clades of similarity (line 619). These observations or findings should be stated in a more specific and quantitative way.
Some minor points are also indicated below.
1. The full names of abbreviations, such as PCA and ns in the abstract and the main text, should be provided during their first use.
2. The standard abbreviations for acetyl and methylamine are Ac and MMA, respectively, instead of ACE and NME, in organic chemistry.
3. On p. 19, line 552, supplementary figure # should be corrected.
Overall, this manuscript needs major revision. It is not recommended for publication in Viruses at its current form.

English needs to be extensively revised by an English native speaker.
